# AntiBP3: A Method for Predicting Antibacterial Peptides against Gram-Positive/Negative/Variable Bacteria

**DOI:** 10.3390/antibiotics13020168

**Published:** 2024-02-08

**Authors:** Nisha Bajiya, Shubham Choudhury, Anjali Dhall, Gajendra P. S. Raghava

**Affiliations:** Department of Computational Biology, Indraprastha Institute of Information Technology, Okhla Phase 3, New Delhi 110020, India; nishab@iiitd.ac.in (N.B.); shubhamc@iiitd.ac.in (S.C.); anjalid@iiitd.ac.in (A.D.)

**Keywords:** antibacterial peptides, fastText Embeddings, therapeutic peptides, gram-positive/negative/variable bacteria, machine learning, deep learning, similarity search

## Abstract

Most of the existing methods developed for predicting antibacterial peptides (ABPs) are mostly designed to target either gram-positive or gram-negative bacteria. In this study, we describe a method that allows us to predict ABPs against gram-positive, gram-negative, and gram-variable bacteria. Firstly, we developed an alignment-based approach using BLAST to identify ABPs and achieved poor sensitivity. Secondly, we employed a motif-based approach to predict ABPs and obtained high precision with low sensitivity. To address the issue of poor sensitivity, we developed alignment-free methods for predicting ABPs using machine/deep learning techniques. In the case of alignment-free methods, we utilized a wide range of peptide features that include different types of composition, binary profiles of terminal residues, and fastText word embedding. In this study, a five-fold cross-validation technique has been used to build machine/deep learning models on training datasets. These models were evaluated on an independent dataset with no common peptide between training and independent datasets. Our machine learning-based model developed using the amino acid binary profile of terminal residues achieved maximum AUC 0.93, 0.98, and 0.94 for gram-positive, gram-negative, and gram-variable bacteria, respectively, on an independent dataset. Our method performs better than existing methods when compared with existing approaches on an independent dataset. A user-friendly web server, standalone package and pip package have been developed to facilitate peptide-based therapeutics.

## 1. Introduction

Since their initial discovery, the widespread use of antibiotics has resulted in the emergence of drug-resistant strains of pathogenic bacteria [1]. Peptide-based therapeutics have gained considerable attention in the last few decades to manage drug-resistant strains of bacteria. Antibacterial peptides (ABPs) are short oligopeptides of between 5 and 100 amino acid residues [2,3,4]. They are distinguished by their cationic nature and enriched with certain amino acids such as Arginine and Lysine. These peptides have amphipathic properties, which aid in their incorporation into pathogen cell membranes (Figure 1) [5,6]. Additionally, ABPs can employ intracellular modes of action, such as modulating enzymatic activity, protein degradation and synthesis, and nucleic acid synthesis. This multifaceted mode of action provides an advantage over traditional antibiotics in terms of reduced development of resistance [2,4,7,8].

Though experimental methods are highly accurate, they are not suitable for high throughput, as these are time-consuming, labour-intensive, and expensive (Figure 1) [9,10,11,12]. To address this challenge, numerous in silico methods have been developed for predicting antibacterial peptides. In 2007, Lata et al. developed a method of AntiBP using machine learning (ML) techniques to classify antibacterial and non-antibacterial peptides [13]. Since then, numerous prediction models have been developed that include AmpGram, AI4AMP, AMPfun, and AMPScanner [14,15,16,17,18,19,20,21,22,23]. In addition, attempts have been made to develop class-specific prediction servers, like AntiBP2, for predicting the source of antibacterial peptides [24,25,26].

One of the most significant disadvantages of present approaches is their inability to address all bacterial types, notably gram-(indeterminate/variable) bacteria, which cannot be detected by gram-staining methods. Furthermore, the models are often trained on limited data that are no longer a representation of the current state of the field. In order to complement the existing method, a conscious and systematic effort was undertaken to address the aforementioned shortcomings. The focus of this study is to build prediction models capable of identifying antibacterial peptides for all categories of bacteria, including gram-variable strains, by leveraging large and diverse datasets.

## 2. Results

### 2.1. Compositional Analysis

We calculated the composition of amino acids for three groups of ABP training datasets. The graph in Figure 2 depicts a comparison of average compositional analyses for the three groups of ABPs. In the case of the gram-positive (GP) ABPs dataset, amino acids such as Leu, Lys, Arg, Cys, and Trp have a higher average composition compared to non-ABPs. Regarding the gram-negative (GN) ABPs, the average composition of residues like Leu, Lys, Arg, Trp, and Pro is higher. Similarly, for the gram-variable (GV) ABPs, the amino acids Leu, Lys, Arg, and Trp have a higher composition, respectively, compared to non-ABPs. It is important to note that different types of ABPs (e.g., GP, GN, GV) have different compositions for different amino acids; for example, see the composition of residues Cys, Lys, and Arg. These observations indicate that each group of ABPs has different properties.

### 2.2. Positional Conservation Analysis

We conducted positioning analyses for our three sets of ABPs to determine preference patterns. As shown in Figure 3, residues like Lys, Arg, Ala, Leu, and Phe predominate in the first position of GP ABPs at the N-terminus, whereas Leu, Arg, and Trp are commonly found in the second position, while the C-terminus of GP ABPs is dominated by residues Gly, Trp, Arg, and Lys. At the N-terminus of GN ABPs, residues like Lys, Val, Arg, Leu, and Phe predominated in the first position, whereas Lys, Leu, Arg, Trp, and Phe were commonly found in the second position. Similarly, at the C-terminus, specific residues are favoured; for example, residues Lys, Arg, Leu, and Trp are present at most locations. In the case of GV ABPs at the N-terminus, the first position is favoured by Gly, Phe, Lys, Arg, Ile, and Leu, whereas Leu, Trp, Lys, Ile, Arg, and Phe were often found in the second position. Meanwhile, the C-terminus is dominated by the residues Lys, Leu, Arg, Trp, Phe, Ile, and Gly. Overall, when the amino acid composition of ABP and non-ABP is compared, positively charged Lys and Arg are predominant in antibacterial peptides (see Figure 3). Similarly, Gly and Leu propensity are high in ABPs because they are hydrophobic in nature, which aids in bacterial lipid membrane integration.

### 2.3. Performance of BLAST-Search

We utilised BLAST to query the ABPs of the validation dataset against sequences in training datasets. Various e-values were used to determine the optimal e-value at which the BLAST module performed the best. Hits were obtained from the total number of peptides in the validation set (372 for GP ABPs, 582 for GN ABPs and 3594 for GV ABPs). Results for e-value for three groups of ABPs from 10^−20^ to 10^3^ for GP ABPs, GN ABPs, and GV ABPs are shown below in Table 1. Though the number of hits grows with increasing e-values, performance suffers as it allows incorrect hits.

### 2.4. Performance of Motif-Based Approach

We discovered GP, GN, and GV-associated motifs from training datasets of three ABP groups using MERCI. The training dataset was used to identify discriminatory patterns for each group, and the existence of these motifs was then utilized to give group labels to each peptide sequence in the validation dataset. The number of distinct motifs at three different frequencies in the training set and the proportion of accurate motifs discovered in the validation dataset for each group are shown in Table 2.

The motifs like ‘YGN, YGNG, YGNGV, and NNG’ are solely present in GP antibacterial peptides. Alternatively, the motifs ‘RPPR, PRPR, RRIY, and LPRP’ are exclusively found in GN antibacterial peptides. Similarly, motifs such as ‘KKLLKK, RIVQ, SKVF, and RIVQR’ are only found in GV antibacterial peptides. The detailed results of the top 10 motifs in GP, GN, and GV ABPs at fp10 are provided in Table 3. The total number of occurrences of the top 10 exclusive motifs for GP, GN, and GV is 373, 682, and 509, while the coverage sequences are 90, 107, and 275, respectively.

### 2.5. Machine Learning-Based Models

We created three independent prediction models on training datasets of three groups of ABPs, utilizing different ML classifiers such as Random Forest (RF), Decision Tree (DT), Gaussian Naive Bayes (GNB), Logistic Regression (LR), Support Vector Classifier (SVC), K-Nearest Neighbour (KNN), and Extra Tree (ET), and analyzed the performance of these models on different feature sets.

#### 2.5.1. Performance of Composition-Based Models

In this section, we computed the performance of 17 distinct compositional features. We discovered that ET-based classifiers outperformed all other classifiers by comparing the AUC-ROC curves (see Appendix A). Using AAC-based features, we achieved maximum performance on the validation set with an AUC of 0.94 and MCC of 0.75 for GP ABPs. Similarly, employing PCP-based features on the validation set, we obtained comparable results (i.e., AUC = 0.94 and MCC = 0.72). Other composition-based features perform well across all three datasets. In the case of GN ABPs, we attain a maximum AUC of 0.98 and MCC of 0.86 using AAC-based features. On the validation set of GN ABPs, all other features likewise operate admirably. While using AAC-based features in the case of GV ABPs, we achieved an AUC of 0.95 and MCC of 0.74 on the validation set. Here, APAAC also performs better on the validation set, whereas SOC, BTC, and SEP underperform. The complete training and testing dataset results on all features for GP, GN, and GV ABPs are provided in Table 4, and the rest of the results are provided in Appendix A.

#### 2.5.2. Performance of Binary Profile-Based Features

We also built machine learning-based models using binary profiles of N, C, and NC terminals to incorporate information about frequency as well as the order of residues (Table 5). For different binary profile features in the GP ABPs dataset, the RF, SVC, and ET models outperform all other classifiers. On the training set, AAB performed best for the NC-terminal, with AUC and MCC of 0.91 and 0.64, respectively, and a similar trend was found in the validation set, with AUC and MCC of 0.93 and 0.66, respectively. (The complete result is provided in Appendix A.) Similarly, for the GN ABPs dataset, the ET and SVC models outperform all other classifiers. Here, all the binary features perform quite well and show the highest AUC of 0.98 on the validation set for the NC-terminal with an MCC of 0.86. For the GV ABPs dataset, the SVC and RF models perform best among all other classifiers for the binary profile features. A similar trend was found here also, as the NC-terminal performed best for binary features on the training set, with an AUC of 0.92, and also performed well on the validation set, with an AUC of 0.94. Therefore, the models developed using the binary profile features of the NC-terminal perform superior to individual N and C termini, suggesting that both the N and C-terminals are crucial in distinguishing gram-positive ABPs from non-ABPs.

#### 2.5.3. Performance of Word Embedding Features

In this feature extraction method, we wanted to assess the efficiency of different segmentation sizes and the n-gram mixture employed. We evaluated the overall performance of several classifiers from each ABP group in both training and testing tests. The outcomes of the classifiers are shown in Table 6. Among the four segmentation sizes, the feature type corresponding to biological words of length one and two contributed to the best overall average performance for all groups using an ET-based model. This demonstrates the significance of individual amino acid residues in an antibacterial peptide sequence. The complete results are provided in Appendix A.

### 2.6. Deep Learning-Based Models

We built prediction models on training datasets of GP ABPs, utilizing different deep learning (DL) classifiers such as artificial neural networks (ANN), recurrent neural networks (RNN), convolutional neural networks (CNN) and long short-term memory (LSTM), and analyzed their performance on three different feature sets. By comparing the AUC-ROC curves on different sets of features of GP APBs, we discovered that ML-based classifiers outperformed all DL-classifiers (see Figure 4, Figure 5 and Figure 6).

The performance of all ML and DL-based models on the top-performing amino acid binary feature of the NC-terminal of GP ABPs is shown in Table 7. Among all the ML models, RF outperforms with an AUC and MCC of 0.93 and 0.71, while in the case of DL models, ANN performs better with an AUC and MCC of 0.89 and 0.62. The overall performance of all DL models on compositional, binary, and word embedding features of GP ABPs are provided in Appendix A, respectively. We found that the DL models performed significantly less effectively than the ML models on all feature sets of GP ABPs. The one possible reason for observed lower performance of DL algorithms in this context might be attributed to the comparatively limited dataset size, as DL performs better on bigger datasets. Therefore, in order to reduce computational time and effort, we choose not to run the DL models on the other two ABPs, i.e., GN and GV. As a result, we did not proceed with the DL-based models in our further research.

### 2.7. Performance of Cross-Prediction

We evaluated the performance of each binary features-based ML model on both training and testing datasets (see Appendix A). In this study, we perform cross-prediction of our best-performing models, i.e., RF-based, ET-based, and SVC-based models for GP, GN, and GV ABPs, respectively, on different validation datasets, and the best performance is highlighted in bold, shown in Table 8. The models perform best with their own datasets and are significantly poorer when making predictions with different groups of validation datasets. The AUC of the models falls with different sets of validation data, such as in the case of gram-positive ABPs, where the model trained with GP ABPs has an AUC of 0.93 on the GP validation dataset, drops to 0.89 and 0.91 on the GN and GV validation datasets. In the case of GN ABPs, models attain an AUC of 0.98 on their own validation dataset and drop to 0.85 and 0.93 on the GP and GV validation datasets, respectively. Similarly, the AUC of the GV ABPs model on its own validation dataset is 0.94, which drops to 0.88 and 0.93 for GP and GN validation datasets. This trend indicates the necessity for developing different models for each of the three groups of ABPs. We have also generated a confusion matrix of the best-performing model using the binary feature AAB (NC-terminal), given in Appendix A.

### 2.8. Performance of Hybrid Approaches

From the above results, we have observed that AAB-based features for the NC terminal outperformed for all three groups of ABPs prediction models. Therefore, to construct the final predictions, we coupled BLAST similarity and MOTIF scores with best-performing ML-model scores computed using AAB (NC-terminal) features.

#### 2.8.1. ML-Based Models with BLAST Search

In this study, in order to improve the performance of the individual ML models, we developed three hybrid models using the BLAST + ML technique to categorise ABPs into three categories. We began by using a similarity search (BLAST) to predict positive and negative peptides. We calculated the performance of three hybrid models on validation datasets using the best feature and best model at different e-value cut-offs. Once the ML model has made the forecast, the presence of the hit is utilised to fine-tune the prediction.

In the case of GP ABPs, the performance of the hybrid approach with BLAST slightly increases, more than the individual ML approach with AUC and AUPRC of 0.94 and 0.93, respectively, at an e-value of 10^−20^. However, there is no substantial increase in the performance of the hybrid model using this method in the other two scenarios for GN and GV ABPs, even at the lowest evaluated e-value of 10^−20^. The detailed results of the hybrid model with this BLAST approach are provided in Appendix A. These results indicate that BLAST + ML is not better than ML alone.

#### 2.8.2. ML-Based Models with Motif Approach

Similarly, we constructed hybrid models using the MOTIF-search and ML called Motif+ML. First, we utilised the motif search to predict positive and negative peptides based on whether or not a motif was detected in validation data. The best model’s score is then added to the motif score to calculate the performance of three hybrid models on validation datasets at varying frequencies. However, we observed the same pattern as with the BLAST + ML strategy, with no substantial improvement in the performance of the hybrid model. The detailed results of the hybrid model with this MOTIF approach are provided in Appendix A.

Overall, combining ML-based models with blast and motif approaches does not outperform individual ML-based models.

### 2.9. Comparison with Other Prediction Tools

Our best-performing models were compared to five different ML and DL-based ABP predictors, namely, AMPScanner vr.2, AI4AMP, iAMPpred, AMPfun, and ABP-Finder. Currently, only a few tools, such as ABP-Finder and AMPfun, examine the distinction of antibacterial peptide groups based on gram staining. Other techniques, such as AMPScanner vr2, iAMPpred, and AI4AMP, predict whether or not the peptide has broad antibacterial action. We evaluated the performance of existing methods on our validation datasets. In the case of the two-stage classifier ABP-Finder, the performance was evaluated based on the type of validation set employed, as it considers classes of antibacterial peptides. Predictions of the validation set of GP ABPs are regarded as correct only if designated as gram-positive; otherwise, predictions of gram-negative or both are considered incorrect. Similarly, the accurate estimation for GN ABPs should be gram-negative, whereas predictions given as gram-positive or both are incorrect. In the case of GV ABPs, showing predictions as both are regarded right, whereas either gram-positive or gram-negative prediction is considered incorrect. As we obtained output in binary form, AUC cannot be computed for the ABB-finder. Similarly, the discrepancy in sensitivity and specificity for all other tools is very large for all three sets of ABPs (Table 9). When we ran the validation set on our model, we obtained AUC of 0.93, 0.98, and 0.94 for GP ABPs, GN ABPs, and GV ABPs, respectively, which is greater than any of the abovementioned tools. Because of fewer false positives and the greatest AUC, AntiBP3 is able to attain a balanced sensitivity and specificity.

### 2.10. Implementation of AntiBP3 Server

We integrated our best-performing models in AntiBP3, a user-friendly online server (https://webs.iiitd.edu.in/raghava/antibp3/ (accessed on 20 November 2023)) that predicts antibacterial peptides for different groups of bacteria based on sequence information for each entry. The web server includes five main modules: (i) Predict, (ii) Design, (iii) Protein scan, (iv) BLAST scan, (v) Motif scan, and (vi) Package. The web server (https://webs.iiitd.edu.in/raghava/antibp3 (accessed on 20 November 2023)), python based standalone package (https://webs.iiitd.edu.in/raghava/antibp3/package.php (accessed on 20 November 2023)), GitLab (https://gitlab.com/raghavalab/antibp3 (accessed on 20 November 2023)) and PyPI repository (https://pypi.org/project/antibp3/ (accessed on 20 November 2023)) are freely-available to the community.

## 3. Discussion

ABPs, belonging to the class of antimicrobials, play a crucial role in innate immunity similar to defensins and exhibit potent activity against drug-resistant bacteria [27]. The field of in silico prediction of antibacterial peptides has witnessed the development of various methods. One of the major obstacles faced by researchers is the continuous updating of these models to incorporate the latest information. In 2010, we developed AntiBP2 for predicting ABPs, which was trained on a small dataset (999 ABPs and 999 non-ABPs). Despite it being heavily used and cited by the scientific community, it has not been updated since 2010. This is true with most of the existing methods. In this study, we made a systematic attempt to develop prediction models using the most up-to-date information. Rather than adopting a single generalised model like AntiBP2, here we developed different bacterial group-specific methods for predicting ABPs, allowing us to capture the distinct features and variations within each group. In this study, we created three prediction models to classify antibacterial peptides into three categories of ABP against gram-positive, gram-negative, and gram-variable bacteria [28,29,30].

We developed numerous prediction models using alignment-free ML and DL techniques from sequence-based features, including compositional and binary profiles. We investigated every possible feature in an attempt to predict the various categories of ABPs with high accuracy. We discovered that among all the compositional features-based models, the model created utilizing basic amino acid composition outperformed the models developed using other sophisticated compositional features. In the case of binary-based features, the NC terminus, which considers both amino acid order and composition, performs better than N- or C-terminus-based features concordant with the trend followed by the AntiBP and AntiBP2 method [31,32]. However, fastText features also perform comparably with compositional or binary-based features. By evaluating the ML and DL performance on all these feature sets, we discovered that the ML-based approaches are better than the DL; hence, we didn’t continue with the DL further.

Along with the ML and DL approaches, commonly used alignment-based methods, BLAST and Motif-search, were used to annotate the peptide sequence. However, our hybrid models (BLAST + ML and MOTIF + ML) do not outperform individual ML-based models, which might be owing to the prevalence of false positives even in extreme cases, such as at the lowest e-value of 10^−20^, which reduces the AUC in hybrid techniques. Our method outperformed existing methods on an independent or validation dataset, which is not used to build models. This implies that our approach can more precisely categorise antibacterial peptides into their respective groups.

## 4. Materials and Methods

### 4.1. Dataset

We collected the ABPs from multiple repositories that specialize in antibacterial peptides for gram-positive, gram-negative, and gram-variable bacteria. These public repositories include APD3, AntiBP2, dbAMP 2.0, CAMPR3, DRAMP, and ABP-Finder, as depicted in Figure 7 [12,24,33,34,35,36]. We removed sequences containing non-standard amino acids (BJOUXZ), as well as eliminating sequences shorter than eight and longer than 50 amino acids [37]. Our final dataset consists of 11,370 unique antibacterial peptides belonging to three groups: 930 peptides for GP bacteria, 1455 peptides for GN bacteria, and 8985 peptides for GV bacteria. To create non-antibacterial peptides (non-ABPs) for the negative dataset, we followed a similar approach to that used in AntiBP and AntiBP2 [13,24,38]. This search yielded 11,257 sequences, which were then integrated with the negative dataset used in the ABP-Finder. We then removed the duplicate sequences, resulting in a final set of 10,000 unique sequences called non-ABPs. This non-ABP data was then randomly split according to different classes of ABP training and validation sets, in order to create a balanced dataset.

We divide the entire data into training and validation datasets, where the training dataset contains 80% of the data, and the validation dataset contains 20% of the data (Figure 7). In the case of GP, the training dataset comprises 744 ABP and 744 non-ABP, whereas the validation dataset has 186 ABP and 186 non-ABP. Similarly, the training dataset for GN contains 1164 ABP and 1164 non-ABP, and the validation dataset includes 291 ABP and 291 non-ABP. Likewise, the training dataset for GV contains 7188 ABP and 7188 non-ABP, whereas the validation dataset contains 1797 ABP and 1797 non-ABP. The validation dataset was not utilized either for training or testing the method and has been used to assess the performance of the prediction models.

### 4.2. Overall Workflow

The complete workflow of the current study is illustrated in Figure 8.

### 4.3. Two Sample Logo

We used the Two Sample Logo (TSL) to construct sequence logos for the three categories of ABPs in both the positive and negative datasets [39]. In the sequence logos, the *x*-axis depicts the amino acid residues in the generated sequence logos. At the same time, the positive *y*-axis displays the bit-score for the enriched residues, whereas the negative *y*-axis displays the depleted residues in the given peptide sequences, reflecting the relevance of a single residue at a given location. TSL, on the other hand, accepts peptide sequences as a fixed-length vector for input. Therefore, we combined eight amino acids from the N-terminus (beginning) and eight residues from the C-terminus (end) for each peptide sequence, to create a fixed-length vector of sixteen residues corresponding to the shortest peptide length in our dataset, which is eight residues.

### 4.4. Sequence Alignment Method

In this study, we used an alignment-based approach to categorise ABPs into GP, GN, and GV based on similarity. We utilized the NCBI-BLAST+ version 2.13.0+ (blastp suite) to implement a similarity search approach [40,41]. We created a specialized ABP database containing the respective ABP sequences and a non-ABP database containing non-ABP sequences for each group set (GP, GN, and GV) using the makeblastdb suite. Finally, the validation dataset was then queried against the custom databases using the BLASTP suite. The peptides were classified based on the presence of the most significant hit in either the respective ABPs or the non-ABPs databases. If the top BLAST result was an ABPs peptide from a certain group, the query sequence was classified as a member of that group. On the other hand, if the top hit was from the non-ABP database, the peptide was classified as a non-ABP sequence. We conducted the BLAST with several e-value cut-offs ranging from 10^−20^ to 10^3^ to discover the optimal value of the e-value threshold.

### 4.5. Motif Search

In our study, we investigated the motif-based strategy for identifying conserved motifs in ABPs [42,43]. We employed Motif—EmeRging and with Classes—Identification (MERCI) [44], a tool for identifying conserved motifs that use Perl script. In MERCI, both the positive and negative datasets were given as input, but it gave only the motifs for the positive sequences at a time. The algorithm then found ungapped patterns/motifs with a motif occurrence frequency of 10, 20, and 30 (fp10, fp20, and fp30) that may successfully differentiate between positive and negative samples for the particular ABP group. Here, the parameter fp corresponds to a criterion that determines how prevalent a motif (a pattern in the peptide sequences) should be in order to be considered relevant i.e., motifs with occurrences beyond these thresholds are retained. This will exclude motifs that appear less frequently than the specified threshold, which allows the algorithm to focus on more significant and frequently occurring motifs in the dataset. The selected numbers for fp are arbitrary; additionally, on increasing the frequency threshold, the coverage decreases. Thus, we obtained three distinct sets of motifs for each ABP group and computed the overall motif coverage.

### 4.6. Feature Generation

In the current work, we estimated various features utilising peptide sequence information. We utilised the Pfeature [45] standalone software to compute composition-based and binary profile-based features of our datasets and developed a prediction model for each of them.

#### 4.6.1. Compositional Features

We calculated seventeen different types of descriptors/features, including AAC (amino acid composition), APAAC (amphiphilic pseudo amino acid composition), DDR (distance distribution of residue), DPC (di-peptide composition), QSO (quasi-sequence order), PCP (physico-chemical properties composition), PAAC (pseudo amino acid composition), RRI (residue repeat Information), SPC (Shannon entropy of physicochemical properties), ATC (atomic composition), BTC (bond type composition), CTC (conjoint triad descriptors), AAI (amino acid index), PRI (property repeats index), SEP (Shannon entropy of a protein), SER (Shannon entropy of a residue), SOC (sequence order coupling number).

#### 4.6.2. Binary Profile Features

In order to compute binary profiles of ABPs and non-ABPs, the length of variables should be fixed. The minimum length of peptides in our dataset is eight; hence, we constructed binary profiles for N8, C8, and combined terminal residues N8C8 after computing the fixed length patterns. Similarly, for each peptide sequence in both positive and negative datasets, we generated four different types of binary profile features for N-, C-, and NC-terminals, including AAB (amino acid-based binary profile), DPB (dipeptide-based binary profile), PCB (physico-chemical properties based binary profile), AIB (amino acid indices-based binary profile).

#### 4.6.3. Word Embedding

Word embedding is a strong natural language processing technique that includes encoding words as vectors in a high-dimensional space based on their contextual information [46]. The fastText method was used as our word-embedding technique [47,48]. The peptides were converted into n-grams of varying sizes, such as 1-g (individual words), 2-g (pairs of nearby words), 3-g (triplets of adjacent words), and a mixture of 2-g and 3-g. We next trained the fastText model on the pre-processed training dataset, which generates word-embedding vectors with a fixed dimension (often several hundred) [49]. Finally, we used these feature vectors to train our model to accomplish the necessary classification or prediction job.

### 4.7. Machine Learning Algorithms

Several machine learning methods were used to create the three classification models in our study. Random forest (RF), decision tree (DT), gaussian naive Bayes (GNB), logistic regression (LR), support vector classifier (SVC), k-nearest neighbour (kNN), and extra tree (ET) are among these methods [50,51,52,53,54,55,56]. To build these classifiers, we used the Scikit-learn package, a prominent Python library for machine learning [57].

### 4.8. Deep Learning Algorithms

DL is a method that uses patterns with several hidden layers to learn hierarchical data representations. Initial layers capture low-level properties, which are then combined with succeeding layers to represent the overall data comprehensively [58]. In this study, varied deep learning methods were implemented to perform the classification tasks. Artificial neural network (ANN) [59], recurrent neural network (RNN) [60], convolutional neural network (CNN) [61] and long short-term memory(LSTM) [62], were among these methods. To build the DL-based models, we have used the Keras framework [63] and at the back-end we used the TensorFlow library [64].

### 4.9. Cross-Validation Techniques

We employed both internal and external validation strategies to evaluate the performance of our models. For internal validation, we used the stratified five-fold cross-validation approach, which helps to reduce biases and overfitting. The training dataset was randomly split into five equal sets, each with a comparable number of ABPs and non-ABPs. Four of the five sets were utilised for training the models, while the fifth set was used for testing. This method was performed five times, with each set acting as the test set once, ensuring robust evaluation across multiple iterations [65]. The model’s performance was then assessed using the average performance throughout the five test sets. The performance of the best model generated using the training dataset was evaluated using the 20% independent validation dataset.

### 4.10. Hybrid or Ensemble Approach

In our study, we also used a hybrid or ensemble technique. The following two hybrid techniques were applied in this case: (i) the alignment-based method (BLAST) was combined with the alignment-free approach (ML-based prediction), and (ii) the motif-based approach (MERCI) was combined with ML-based prediction. We adopted the same approach as used in previous studies to combine alignment-based and alignment-free approaches [41,66,67].

### 4.11. Performance Evaluation

The performance of several models was tested using conventional performance assessment measures. We calculated both threshold-dependent metrics (such as sensitivity, specificity, accuracy, Matthews correlation coefficient (MCC), and F1-score), and independent parameters such as area under receiver operating characteristics (AUCROC) and area under the precision-recall curve (AUPRC). The evaluation parameter formulas are shown in the following Equations (1)–(5):(1)$$Sensitivity\left(Sn\right)=\frac{T_{P}}{T_{P}+F_{N}}$$
(2)$$\;Specificity\left(Sp\right)=\frac{T_{N}}{T_{N}+F_{P}}\;$$
(3)$$\;\;Accuracy\left(Acc\right)=\frac{T_{P}+T_{N}}{T_{P}+T_{N}+F_{P}+F_{N}}$$
(4)$$MCC=\frac{\left(T_{P}\;*\;T_{N}\right)-(F_{P}\;*\;F_{N})}{\sqrt{\left(T_{P}+F_{P}\right)\left(T_{P}+F_{N}\right)(T_{N}+F_{P})(T_{N}+F_{N})}}$$
(5)$$F1=\frac{{2T}_{P}}{{2T}_{P}+F_{P}+F_{N}}$$
where true positive (T_P_) and true negative (T_N_) are successfully predicted ABPs and non-ABPs, while false positive (F_P_) and false negative (F_N_) are incorrectly predicted ABPs and non-ABPs. MCC penalized over and under prediction. The harmonic means of the accuracy and recall scores are used to determine the F1-score [43].

## 5. Conclusions

Developing in-silico prediction tools for designing and synthesizing novel ABPs saves a significant amount of time and resources in screening peptide libraries. Separate prediction algorithms are required to address the relevance of the AMP’s particular activity, specifically antibacterial activity, source organism, and so on. In this study, we analyse different ML models that leverage distinct peptide properties for training. However, we demonstrated that all feature generation methods performed similarly. This highlighted the point that the crucial information for distinguishing antibacterial peptides can be captured solely from amino-acid sequence data without the need for sophisticated feature extraction methods. This will simplify and improve the interpretability of the modelling procedures. This study will aid in developing better and more effective antibacterial peptides against resistant strains of all bacterial classes in the future.

## Figures and Tables

**Figure 1 antibiotics-13-00168-f001:**
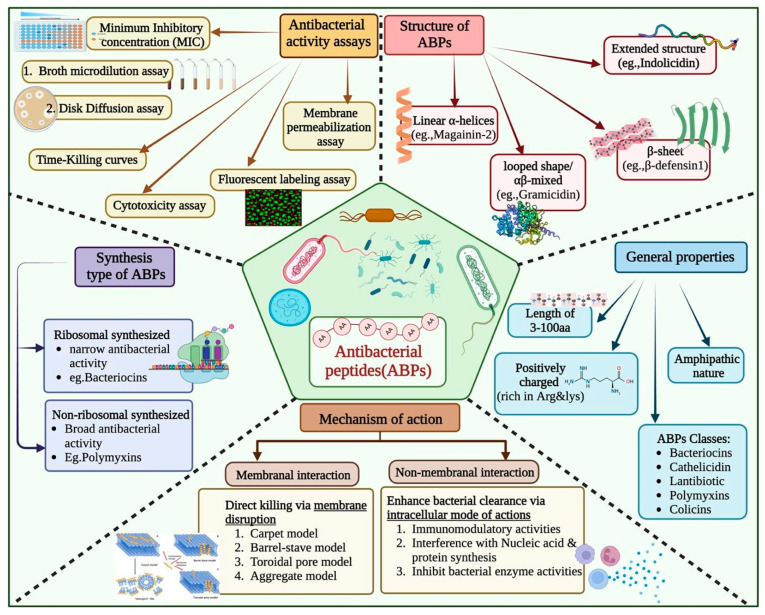
Overview of Antibacterial Peptides: Properties, Mechanisms, Assays, Structures, and Synthesis Type.

**Figure 2 antibiotics-13-00168-f002:**
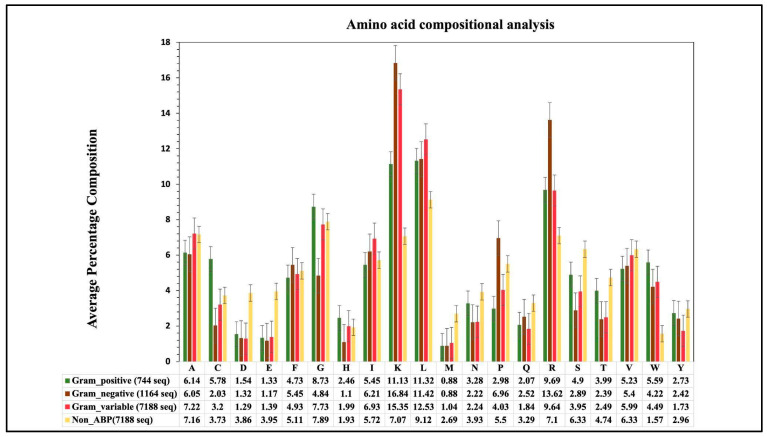
Average amino-acid composition of ABPs and non-ABPs for gram-positive, negative and variable bacteria.

**Figure 3 antibiotics-13-00168-f003:**
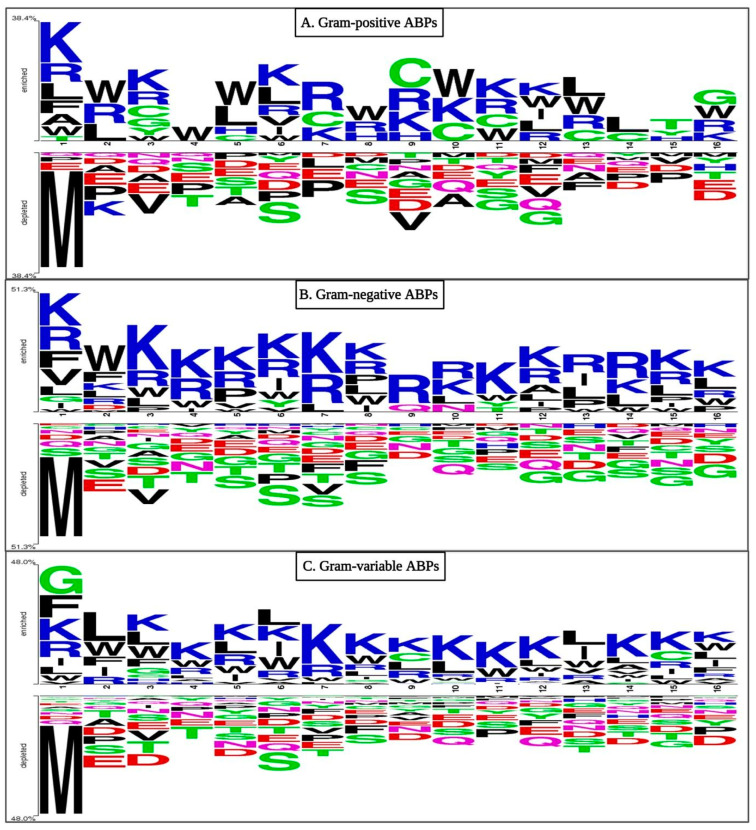
Two sample logo (TSL) representations of ABPs ((**A**) Gram-positive, (**B**) Gram-negative, and (**C**) Gram-variable bacteria) showing preferred positions for amino acids. Here, the first eight residues belong to the N-terminus, while the last eight residues belong to the C-terminus of the peptide.

**Figure 4 antibiotics-13-00168-f004:**
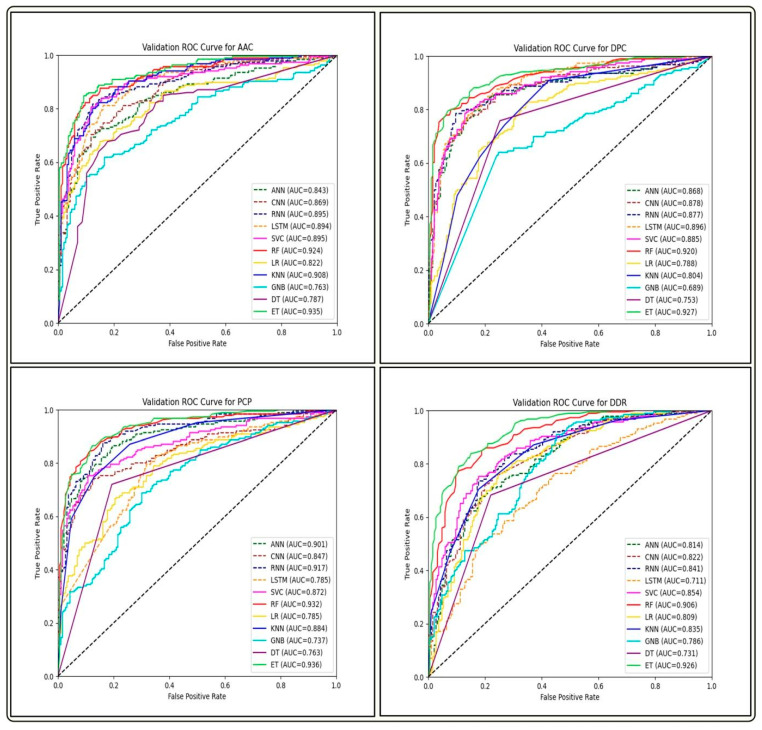
Comparison of AUC-ROC performance across various ML and DL models using four compositional-based features on the validation set of GP ABPs.

**Figure 5 antibiotics-13-00168-f005:**
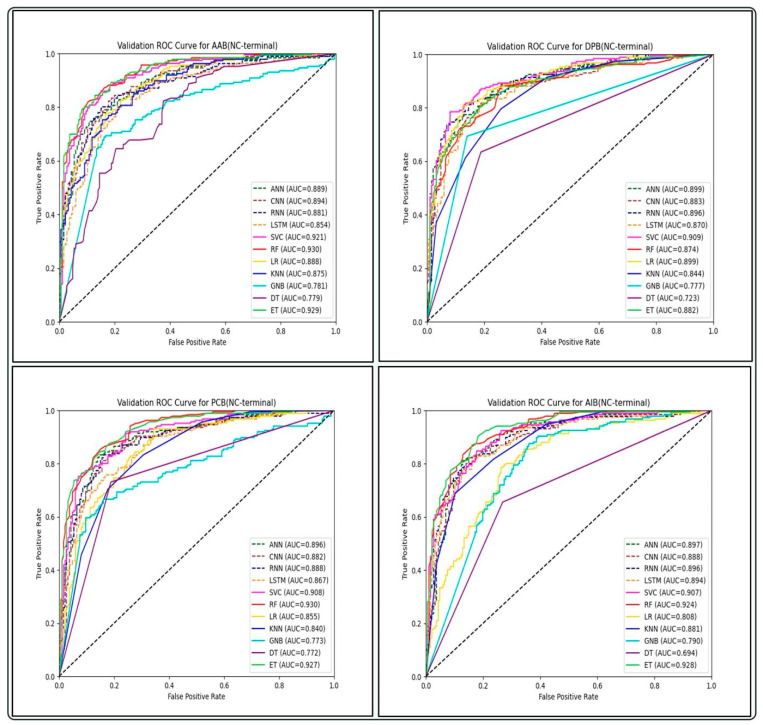
Comparison of AUC-ROC performance across various ML and DL models using binary profile-based features of NC-terminal on the validation set of GP ABPs.

**Figure 6 antibiotics-13-00168-f006:**
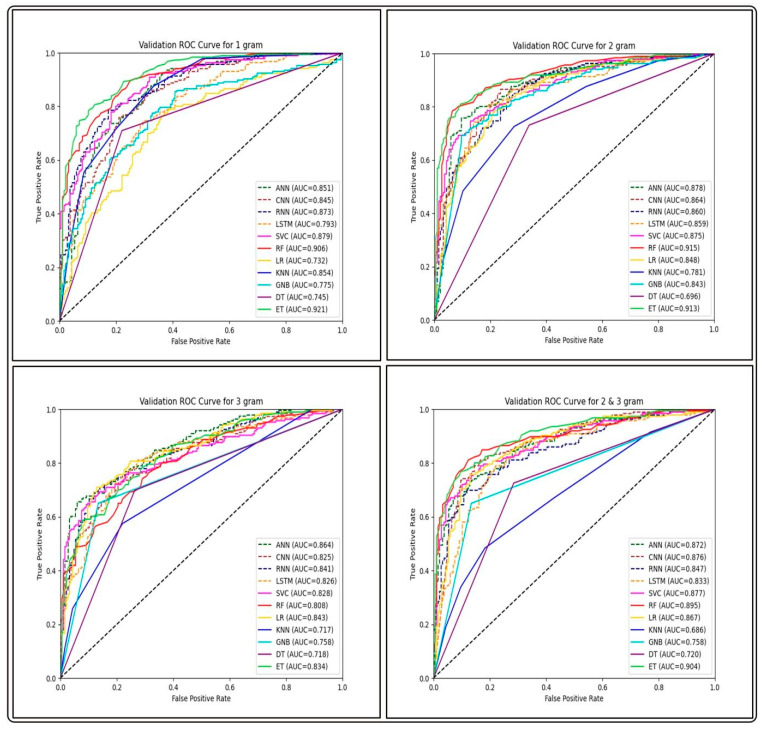
Comparison of AUC-ROC performance across various ML and DL models using FastText-based features on the validation set of GP ABPs.

**Figure 7 antibiotics-13-00168-f007:**
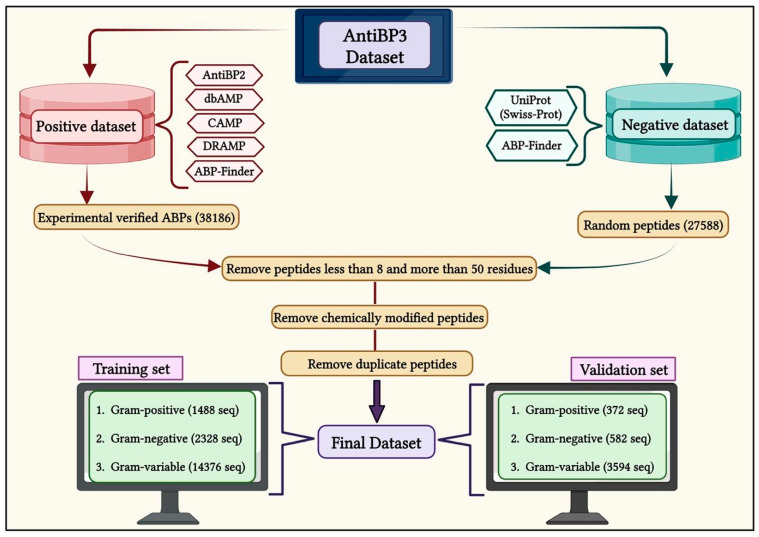
A flow diagram showing the process of creating AntiBP3 datasets for gram-positive, gram-negative, and gram-variable bacteria.

**Figure 8 antibiotics-13-00168-f008:**
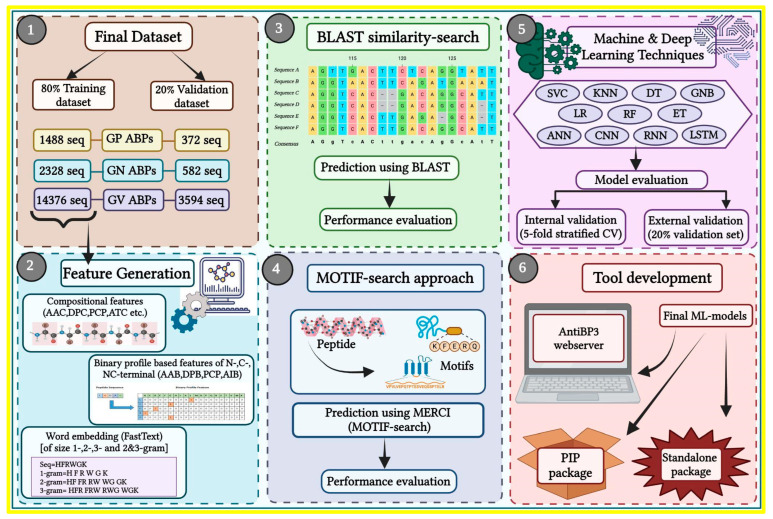
Overall study architecture from feature generation to web server development. The numbers indicate the order in which the workflow is performed.

**Table 1 antibiotics-13-00168-t001:** Comparison of performance among BLAST modules at different e-values for GP, GN, and GV ABPs.

e-Value	Hits			Correct Hits			Incorrect Hits			Percentage of Correct Hits		
	GP	GN	GV	GP	GN	GV	GP	GN	GV	GP	GN	GV
10^−20^	39	30	470	39	28	450	0	2	20	100	93	96
10^−10^	110	138	1188	104	133	1042	6	5	146	95	96	88
10^−06^	150	234	1516	141	226	1301	9	8	215	94	97	86
0.0001	173	262	1730	162	253	1477	11	9	253	94	97	85
0.001	195	284	1925	181	275	1649	14	9	276	93	97	86
0.01	211	326	2114	196	315	1809	15	11	305	93	97	86
0.1	230	362	2326	211	349	1992	19	13	334	92	96	86
1	272	416	2608	242	395	2228	30	21	380	89	95	85
10	329	506	3222	268	462	2658	61	44	564	81	91	82
50	357	552	3488	286	489	2840	71	63	648	80	89	81
100	359	557	3537	289	492	2869	70	65	668	81	88	81
200	361	564	3560	290	498	2884	71	66	676	80	88	81
1000	366	571	3584	295	500	2902	71	71	682	81	88	81

GP: gram-positive, GN: gram-negative, GV: gram-variable.

**Table 2 antibiotics-13-00168-t002:** Number of unique motifs found in GP, GN, and GV ABPs at three different frequencies in the training and validation dataset.

	Training Set				Validation Set		
Category of ABPs	Frequency of a Motif	Number of Motifs	Total Occurrence	Coverage (No. of Sequence)	Test File Coverage (No. of Sequence)	Correct Hits	Percentage of Correct Hits
Gram-positive	fp10	329	6105	285	44	35	79.55%
	fp20	126		122	19	16	84.21%
	fp30	14		92	14	12	85.71%
Gram-negative	fp10	778	14,136	735	233	206	88%
	fp20	217		455	152	142	93%
	fp30	81		302	117	113	97%
Gram-variable	fp10	3079	47,627	3268	774	665	86%
	fp20	651		1585	343	321	94%
	fp30	121		826	168	163	97%

**Table 3 antibiotics-13-00168-t003:** Top 10 Motifs exclusively present in GP, GN, and GV antibacterial peptide sequences at fp10.

S. No.	Gram-Positive		Gram-Negative		Gram-Variable	
	Motifs	Hits (No. of Sequences)	Motifs	Hits (No. of Sequences)	Motifs	Hits (No. of Sequences)
1	YGN	47	RPPR	78	KKLLKK	63
2	YGNG	47	PRPR	74	RIVQ	54
3	NNG	44	RRIY	71	SKVF	52
4	YGNGV	38	LPRP	67	RIVQR	51
5	YYGN	36	IYN	67	IVQRI	50
6	YYGNG	36	IYNR	66	KRIVQ	49
7	VDW	34	PRRI	65	VVIR	48
8	NGLP	31	PRRIY	65	RWWR	48
9	PTGL	30	RIYN	65	DFLR	47
10	RCRV	30	RRIYN	64	RIVQRI	47

**Table 4 antibiotics-13-00168-t004:** The performance of extra tree models developed using 17 types of composition-based features on training and validation datasets, evaluated on GP, GN, and GV ABPs.

Feature Type	Training Set						Validation Set					
	GP		GN		GV		GP		GN		GV	
	AUC	MCC	AUC	MCC	AUC	MCC	AUC	MCC	AUC	MCC	AUC	MCC
AAC	0.93	0.71	0.96	0.80	0.93	0.71	0.94	0.75	0.98	0.86	0.95	0.74
DPC	0.91	0.68	0.95	0.77	0.92	0.70	0.93	0.72	0.98	0.86	0.92	0.70
ATC	0.84	0.52	0.91	0.69	0.86	0.58	0.84	0.51	0.95	0.78	0.89	0.62
BTC	0.75	0.36	0.79	0.43	0.76	0.38	0.70	0.29	0.84	0.51	0.76	0.37
CTC	0.88	0.59	0.94	0.75	0.90	0.68	0.90	0.68	0.97	0.82	0.91	0.71
PCP	0.91	0.64	0.95	0.76	0.91	0.67	0.94	0.72	0.97	0.84	0.93	0.72
AAI	0.91	0.67	0.95	0.77	0.91	0.68	0.92	0.71	0.97	0.86	0.92	0.73
RRI	0.89	0.61	0.94	0.75	0.91	0.68	0.91	0.66	0.97	0.82	0.92	0.69
PRI	0.87	0.58	0.94	0.72	0.90	0.66	0.88	0.63	0.97	0.80	0.91	0.67
DDR	0.91	0.65	0.94	0.73	0.91	0.69	0.93	0.67	0.97	0.80	0.92	0.72
SEP	0.61	0.18	0.72	0.37	0.74	0.35	0.60	0.16	0.80	0.44	0.79	0.45
SER	0.92	0.67	0.96	0.78	0.92	0.71	0.92	0.69	0.97	0.86	0.93	0.72
SPC	0.90	0.64	0.94	0.76	0.90	0.67	0.92	0.71	0.96	0.81	0.92	0.71
PAAC	0.93	0.70	0.96	0.80	0.92	0.71	0.93	0.72	0.97	0.87	0.93	0.74
APAAC	0.93	0.70	0.96	0.79	0.92	0.71	0.93	0.72	0.98	0.87	0.95	0.75
QSO	0.92	0.70	0.96	0.79	0.92	0.71	0.93	0.73	0.97	0.84	0.93	0.73
SOC	0.66	0.20	0.76	0.39	0.61	0.15	0.64	0.27	0.79	0.43	0.64	0.20

MCC: Matthews correlation coefficient, AUC: Area under the receiver operating characteristic curve, AAC: Amino acid composition, APAAC: Amphiphilic pseudo amino acid composition, DDR: Distance distribution of residue, DPC: Di-peptide composition, QSO: Quasi-sequence order, PCP: Physico-chemical properties composition, PAAC: Pseudo amino acid composition, RRI: Residue repeat information, SPC: Shannon entropy of physicochemical properties, ATC: Atomic composition, BTC: Bond type composition, CTC: Conjoint triad descriptors, AAI: Amino acid index, PRI: Property repeats index, SEP: Shannon entropy of a protein, SER: Shannon entropy of a residue, SOC: Sequence order coupling number, GP: gram-positive, GN: gram-negative, GV: gram-variable.

**Table 5 antibiotics-13-00168-t005:** The performance of ML models, developed using binary profiles-based features at different terminals on training and validation datasets, evaluated on GP, GN, and GV ABPs.

Feature Type	Terminal	Training Set						Validation Set					
		AUC	MCC	AUC	MCC	AUC	MCC	AUC	MCC	AUC	MCC	AUC	MCC
**Category of ABP (Model)**		**GP (RF)**		**GN (ET)**		**GV (SVC)**		**GP (RF)**		**GN (ET)**		**GV (SVC)**	
**AAB**	N	0.90	0.62	0.94	0.74	0.92	0.70	0.92	0.67	0.97	0.85	0.93	0.73
	C	0.87	0.59	0.92	0.68	0.89	0.64	0.86	0.55	0.95	0.79	0.89	0.63
	NC	0.91	0.64	0.95	0.75	0.92	0.73	0.93	0.66	0.98	0.86	0.94	0.74
**Category of ABP(Model)**		**GP (SVC)**		**GN (SVC)**		**GV (RF)**		**GP (SVC)**		**GN (SVC)**		**GV (RF)**	
**DPB**	N	0.88	0.59	0.93	0.72	0.48	−0.10	0.89	0.63	0.97	0.80	0.86	0.52
	C	0.85	0.54	0.91	0.67	0.88	0.59	0.85	0.53	0.95	0.77	0.86	0.56
	NC	0.90	0.64	0.94	0.73	0.91	0.67	0.91	0.68	0.98	0.87	0.67	0.32
**Category of** **ABP(Model)**		**GP (ET)**		**GN (ET)**		**GV (SVC)**		**GP (ET)**		**GN (ET)**		**GV (SVC)**	
**AIB**	N	0.89	0.59	0.94	0.74	0.92	0.71	0.91	0.65	0.97	0.85	0.93	0.69
	C	0.88	0.56	0.92	0.69	0.89	0.65	0.88	0.57	0.96	0.76	0.89	0.64
	NC	0.91	0.65	0.95	0.73	0.92	0.73	0.93	0.69	0.98	0.85	0.94	0.73
**Category of ABP (Model)**		**GP (RF)**		**GN (ET)**		**GV (SVC)**		**GP (RF)**		**GN (ET)**		**GV (SVC)**	
**PCB**	N	0.90	0.64	0.94	0.72	0.91	0.70	0.92	0.63	0.97	0.84	0.93	0.73
	C	0.88	0.61	0.91	0.67	0.88	0.62	0.86	0.57	0.94	0.74	0.88	0.63
	NC	0.91	0.64	0.94	0.73	0.92	0.72	0.93	0.70	0.98	0.86	0.94	0.75

MCC: Matthews correlation coefficient, AUC: Area under the receiver operating characteristic curve, AAB: Amino acid-based binary profile, DPB: Dipeptide-based binary profile, PCB: Physico-chemical properties based binary profile, AIB: Amino-acid indices based binary profile, GP: gram-positive, GN: gram-negative, GV: gram-variable, RF: Random forest classifier, ET: Extra-tree classifier, SVC: Support vector classifier.

**Table 6 antibiotics-13-00168-t006:** The performance of best ML models developed using FastText-based features on training and validation datasets, evaluated on GP, GN, and GV ABPs.

Feature (n-Gram Size)	Training Set						Validation Set					
	GP		GN		GV		GP		GN		GV	
	AUC	MCC	AUC	MCC	AUC	MCC	AUC	MCC	AUC	MCC	AUC	MCC
1 g	0.91	0.67	0.95	0.77	0.89	0.64	0.92	0.68	0.97	0.84	0.92	0.71
2 g	0.91	0.66	0.94	0.75	0.92	0.71	0.91	0.71	0.97	0.82	0.93	0.74
3 g	0.87	0.57	0.92	0.70	0.91	0.68	0.84	0.56	0.96	0.82	0.92	0.68
2 g & 3 g Combined	0.91	0.66	0.95	0.77	0.92	0.73	0.90	0.65	0.97	0.82	0.93	0.73

MCC: Matthews correlation coefficient, AUC: Area under the receiver operating characteristic curve. Best ML-model: for GP and GN ABP; 1 g, 2 g, and 2 g & 3 g combined = ET (Extra-tree) classifier, while for GV ABP; 1 g, 2 g, and 2 g & 3 g combined = SVC (Support vector classifier). For 3 g in all three ABPs = LR (Linear regression) performed best.

**Table 7 antibiotics-13-00168-t007:** The comparison of performance of DL and ML models on the validation dataset developed using AAB-based features of the NC -terminal for gram-positive ABP.

Models	Sn	Sp	FPR	Acc	AUC	AUPRC	F1	Kappa	MCC
**ML models**									
DT	64.5	79.6	20.4	72.0	0.78	0.74	0.70	0.44	0.45
RF	80.7	90.3	9.7	85.5	0.93	0.92	0.85	0.71	0.71
LR	72.0	86.0	14.0	79.0	0.89	0.89	0.78	0.58	0.59
KNN	71.0	85.0	15.1	78.0	0.88	0.87	0.76	0.56	0.57
GNB	70.4	79.6	20.4	75.0	0.79	0.80	0.74	0.50	0.50
ET	80.1	89.8	10.2	85.0	0.93	0.92	0.84	0.70	0.70
SVC	76.3	90.9	9.1	83.6	0.92	0.92	0.82	0.67	0.68
**DL models**									
ANN	76.9	85.0	15.1	80.9	0.89	0.89	0.80	0.62	0.62
CNN	75.3	85.5	14.5	80.4	0.89	0.90	0.79	0.61	0.61
RNN	79.6	84.4	15.6	82.0	0.88	0.89	0.82	0.64	0.64
LSTM	68.8	85.0	15.1	76.9	0.85	0.86	0.75	0.54	0.54

Sn: Sensitivity, Sp: Specificity, Acc: Accuracy, MCC: Matthews correlation coefficient, AUC: Area under the receiver operating characteristic curve, AUPRC: Area under the precision-recall curve, AAB: Amino acid-based binary profile, RF: Random forest classifier, SVC: Support vector classifier, ET: Extra-tree classifier, KNN: K-nearest neighbour, GNB: Gaussian naive bayes, LR: Logistic regression, ANN: Artificial neural network, CNN: Convolution neural network, RNN: Recurrent neural network, LSTM: Long short-term memory.

**Table 8 antibiotics-13-00168-t008:** The analysis of the cross-prediction performance of RF, ET, and SVC-based models for GP, GN, and GV ABPs, respectively, using AAB binary feature of NC-terminal on different validation datasets.

Validation Dataset	Performance Measures	Models		
		Gram-Positive	Gram-Negative	Gram-Variable
**Gram-positive** **ABPs**	Sn	80.7	50.0	67.7
	Sp	90.3	89.3	87.6
	Acc	85.5	69.6	77.7
	AUC	0.93	0.85	0.88
	AUPRC	0.92	0.83	0.87
	MCC	0.71	0.43	0.57
**Gram-negative** **ABPs**	Sn	66.0	92.4	91.1
	Sp	89.4	94.9	87.3
	Acc	77.7	93.6	89.2
	AUC	0.89	0.98	0.93
	AUPRC	0.85	0.98	0.92
	MCC	0.57	0.87	0.78
**Gram-variable** **ABPs**	Sn	80.2	76.2	83.9
	Sp	87.4	92.0	90.1
	Acc	83.8	84.1	87.0
	AUC	0.91	0.93	0.94
	AUPRC	0.89	0.93	0.93
	MCC	0.68	0.69	0.74

Sn: Sensitivity, Sp: Specificity, Acc: Accuracy, MCC: Matthews correlation coefficient, AUC: Area under the receiver operating characteristic curve, AUPRC: Area under the precision-recall curve.

**Table 9 antibiotics-13-00168-t009:** The comparison of existing prediction tools with AntiBP3 on the validation dataset to discriminate the three groups of ABPs, i.e., gram-positive, negative and variable.

Method	Algorithm	Sn	Sp	Acc	AUC	AUPRC	MCC
**gram-positive ABPs**							
AMPScanner vr2	CNN & LSTM	84.4	59.7	72.0	0.79	0.75	0.46
AI4AMP	CNN & LSTM	82.8	86.6	84.7	0.90	0.82	0.69
iAMPpred	SVM	79.6	60.8	70.2	0.75	0.71	0.41
AMPfun	SVM	90.3	57.0	73.7	0.83	0.76	0.50
ABP-Finder	RF	8.1	100.0	54.0	-	-	0.21
AntiBP3	RF	80.7	90.3	85.5	0.93	0.92	0.71
**gram-negative ABPs**							
AMPScanner vr2	CNN & LSTM	73.9	45.7	59.8	0.54	0.48	0.20
AI4AMP	CNN & LSTM	67.7	59.8	63.8	0.62	0.55	0.28
iAMPpred	SVM	75.6	44.0	59.8	0.57	0.51	0.21
AMPfun	SVM	73.9	45.7	59.8	0.58	0.52	0.20
ABP-Finder	RF	1.0	100.0	50.5	-	-	0.07
AntiBP3	ET	92.4	94.9	93.6	0.98	0.98	0.87
**gram-variable ABPs**							
AMPScanner vr2	**CNN & LSTM**	90.3	59.0	74.6	0.80	0.74	0.52
AI4AMP	**CNN & LSTM**	91.5	86.2	88.8	0.93	0.88	0.78
iAMPpred	**SVM**	89.8	61.2	75.5	0.78	0.71	0.53
AMPfun	**SVM**	95.5	61.8	78.6	0.87	0.79	0.61
ABP-Finder	**RF**	68.1	79.4	73.7	-	-	0.48
AntiBP3	**SVC**	83.9	90.1	87.0	0.94	0.93	0.74

Sn: Sensitivity, Sp: Specificity, Acc: Accuracy, MCC: Matthews correlation coefficient, AUC: Area under the receiver operating characteristic curve, AUPRC: Area under the precision-recall curve, SVM/SVC: Support vector machine/classifier, RF: Random forest classifier, ET: Extra tree classifier, CNN: Convolution neural network, LSTM: Long short-term memory.

## Data Availability

The datasets constructed for this study can be retrieved from the ‘AntiBP3′ web server at https://webs.iiitd.edu.in/raghava/antibp3/downloads.php (accessed on 20 November 2023). The source code for this study is freely accessible on GitHub and can be found at https://github.com/raghavagps/AntiBP3 (accessed on 20 November 2023).

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
