# Peer review of "AntiBP3: A Method for Predicting Antibacterial Peptides against Gram-Positive/Negative/Variable Bacteria"

_antibiotics, 2024, doi:10.3390/antibiotics13020168_

Round 1

Reviewer 1 Report

Comments and Suggestions for Authors

The manuscript describes a method and a tool for predicting peptides active against gram-positive or gram-negative bacteria. It is a further development of a set of tools reported by the authors in the prior publications and it addresses and important question on AMP predictors with better specificity.

The authors used a diverse set of sequence-based features and a large array of conventional machine learning and deep learning methods. It seems a bit strange that the authors even tried to use deep learning algorithms on their sets, whose sizes are way below of what is usually required for DL, so their underperformance is not surprising. Perhaps, a comment that an insufficient training set size is a possible reason for observed lower performance, might be prudent.

The authors should better explain the details of application of MERCI based motif features. They introduce motifs with “occurrence frequency of 10, 20, and 30 (fp10, fp20, and fp30)”. Are those actual frequencies or frequency thresholds? If it is the former, why these particular numbers were selected? If the numbers are arbitrary, then the explanation of the effect of these numbers on the results would be warranted. Even more important is that the authors use raw counts for the ranking of the motifs found exclusively in one of the peptide classes. This approach can be misleading. The probability of a chance occurrence of a sequence motif is proportional to the relative abundance of individual residues in this motif and reversely proportional to its length. So, for any meaningful and statistically sound comparison of motif content in different classes of peptides, the motif counts should be normalized by amino acid frequencies and motif’s length.

On a technical note, I strongly recommend to round all Sn, Sp, FPR and ACC values in Tables 7, 8, and 9 to one decimal place. The second decimal position is completely meaningless (i.e., it is orders of magnitude smaller than the margin of error) and I can’t see any justification for reporting it. This change would also make the tables more readable.

The paper is well organized and well written. The only noted typo is “significantly poor” instead of “significantly poorer” (p.14).

Author Response

The manuscript describes a method and a tool for predicting peptides active against gram-positive or gram-negative bacteria. It is a further development of a set of tools reported by the authors in the prior publications, and it addresses an important question on AMP predictors with better specificity.

Response: We are thankful to the reviewer for carefully reviewing our manuscript and providing us with useful suggestions. We have tried our best to incorporate all suggestions in the revised version of the manuscript.

Comment 1: The authors used a diverse set of sequence-based features and a large array of conventional machine learning and deep learning methods. It seems a bit strange that the authors even tried to use deep learning algorithms on their sets, whose sizes are way below of what is usually required for DL, so their underperformance is not surprising. Perhaps a comment that an insufficient training set size is a possible reason for observed lower performance might be prudent.

Response: We fully agree with the views of reviewer. As suggested by the reviewer, we have added a comment in our revised manuscript.

Comment 2: The authors should better explain the details of the application of MERCI-based motif features. They introduce motifs with “occurrence frequency of 10, 20, and 30 (fp10, fp20, and fp30)”. Are those actual frequencies or frequency thresholds? If it is the former, why were these particular numbers selected? If the numbers are arbitrary, then the explanation of the effect of these numbers on the results would be warranted. Even more important is that the authors use raw counts for the ranking of the motifs found exclusively in one of the peptide classes. This approach can be misleading. The probability of a chance occurrence of a sequence motif is proportional to the relative abundance of individual residues in this motif and reversely proportional to its length. So, for any meaningful and statistically sound comparison of motif content in different classes of peptides, the motif counts should be normalized by amino acid frequencies and motif’s length.

Response: In this study, we utilized the state-of-the-art software MERCI (PMID 21372086) to identify motifs, which have been extensively employed in previous studies. Our aim was to investigate the potential of a motif-based approach in predicting antibacterial peptides. Specifically, we focused on motifs exclusively present in antibacterial (GP, GN, GV) peptides but absent in non-antibacterial peptides. However, we encountered numerous motifs in the training dataset that appeared only once or in a single sequence, limiting their utility due to poor coverage. To address this issue, we implemented restrictions or thresholds such as fp10, fp20, or fp30. For instance, with fp10, we identified motifs occurring at least 10 times in antibacterial peptides but not in non-antibacterial peptides. Despite this, when we applied these motifs to the test dataset, we observed poor precision (correct/total hits). To enhance precision, we increased the restriction from 10 to 20 or 30, which improved precision at the expense of decreased coverage. Ultimately, we selected the top ten motifs in each class of antibacterial peptides. It is important to note that the improvement of motif detection techniques falls beyond the scope of this paper, as we solely relied on the existing MERCI software. However, in the revised version of the manuscript, we provide a detailed description of our motif detection approach.

Comment 3: On a technical note, I strongly recommend rounding all Sn, Sp, FPR and ACC values in Tables 7, 8, and 9 to one decimal place. The second decimal position is completely meaningless (i.e., it is orders of magnitude smaller than the margin of error), and I can’t see any justification for reporting it. This change would also make the tables more readable.

Response: We are thankful to the reviewer for bringing the mistake to our attention. In Tables 7, 8, and 9, we employed decimal values for AUC, AUPRC, and MCC but not for Sn, Sp, FPR, and ACC, where percentage format is employed; therefore, their magnitudes vary. After getting the comment, we rectified the issue and rounded off the values of Sn, Sp, FPR, and ACC  to one decimal in their respective tables. Additionally, we have also incorporated these changes in the supplementary tables.

Comment 4: The paper is well organized and well written. The only noted typo is “significantly poor” instead of “significantly poorer” (p.14).

Response: We are thankful to the reviewer for their positive feedback on the organization and writing of the paper. After getting the comment, we acknowledged the noted typo error in our manuscript. Subsequently, we have rectified this error in the manuscript and highlighted the same in the revised manuscript.

Reviewer 2 Report

Comments and Suggestions for Authors

This work used in silicon and machine/deep learning based strategies to develop a method for predicting antibacterial peptides against gram-positive/negative/variable bacteria, and further validated this method by cross-prediction and an independent data set. Although this tool is limited by lack of validation using real-world data, it is still a useful tool and would benefit to this field. Therefore, I would recommend to accept it for publication.

Author Response

This work used in silicon and machine/deep learning-based strategies to develop a method for predicting antibacterial peptides against gram-positive/negative/variable bacteria and further validated this method by cross-prediction and an independent data set. Although this tool is limited by lack of validation using real-world data, it is still a useful tool and would benefit to this field. Therefore, I would recommend to accept it for publication.

Response: We are thankful to the reviewer for appreciating our efforts and providing positive feedback.  We will definitely consider the suggestion and strengthen the tool by implementing validation with real-world data in future. Moreover, we have again thoroughly checked and corrected the manuscript, where needed.

Reviewer 3 Report

Comments and Suggestions for Authors

The manuscript of Bajiya et al. reports a novel method for predicting antibacterial peptides (ABPs) against gram-positive/negative/variable bacteria. This novel bioinformatic approach is based on alignment search using BLAST. Subsequently, the authors add a motif-based approach to predict ABPs. In order to test the method, several models were evaluated on an independent dataset with no common peptide between training and independent datasets.

Finally, a user-friendly web server has been developed and made available to researchers.

In my opinion the manuscript is of interest. Furthermore, its use by the scientific community will verify its solidity and potential.

            In my opinion only a careful rereading by an English language expert would be useful.

Author Response

The manuscript of Bajiya et al. reports a novel method for predicting antibacterial peptides (ABPs) against gram-positive/negative/variable bacteria. This novel bioinformatic approach is based on an alignment search using BLAST. Subsequently, the authors add a motif-based approach to predict ABPs. In order to test the method, several models were evaluated on an independent dataset with no common peptide between training and independent datasets.

Finally, a user-friendly web server has been developed and made available to researchers.

In my opinion, the manuscript is of interest. Furthermore, its use by the scientific community will verify its solidity and potential.

In my opinion, only a careful rereading by an English language expert would be useful.

Response: We are thankful to the reviewer for appreciating our efforts and providing valuable suggestions. We are pleased that the reviewer finds the approach interesting and acknowledges the potential utility of the developed web server for the scientific community. Additionally, as per the suggestion provided by the reviewer, we conducted a thorough language review. We have incorporated all suggestions in the revised version of the manuscript.

Round 2

Reviewer 1 Report

Comments and Suggestions for Authors

Authors' response is mostly satisfactory.